# Modulation of the Immune Response to Improve Health and Reduce Foodborne Pathogens in Poultry

**DOI:** 10.3390/microorganisms7030065

**Published:** 2019-02-28

**Authors:** Christina L. Swaggerty, Todd R. Callaway, Michael H. Kogut, Andrea Piva, Ester Grilli

**Affiliations:** 1United States Department of Agriculture/ARS, 2881 F and B Road, College Station, TX 77845, USA; mike.kogut@ars.usda.gov; 2Department of Animal and Dairy Science, University of Georgia, 425 River Road, Athens, GA 30602, USA; todd.callaway@uga.edu; 3Vetagro S.p.A., Via Porro 2, 42124, Reggio Emilia, Italy; ap@vetagro.com (A.P.); eg@vetagro.com (E.G.)

**Keywords:** feed additive, foodborne pathogen, immunomodulation, nutrition, poultry

## Abstract

*Salmonella* and *Campylobacter* are the two leading causes of bacterial-induced foodborne illness in the US. Food production animals including cattle, swine, and chickens are transmission sources for both pathogens. The number of *Salmonella* outbreaks attributed to poultry has decreased. However, the same cannot be said for *Campylobacter* where 50–70% of human cases result from poultry products. The poultry industry selects heavily on performance traits which adversely affects immune competence. Despite increasing demand for poultry, regulations and public outcry resulted in the ban of antibiotic growth promoters, pressuring the industry to find alternatives to manage flock health. One approach is to incorporate a program that naturally enhances/modulates the bird’s immune response. Immunomodulation of the immune system can be achieved using a targeted dietary supplementation and/or feed additive to alter immune function. Science-based modulation of the immune system targets ways to reduce inflammation, boost a weakened response, manage gut health, and provide an alternative approach to prevent disease and control foodborne pathogens when conventional methods are not efficacious or not available. The role of immunomodulation is just one aspect of an integrated, coordinated approach to produce healthy birds that are also safe and wholesome products for consumers.

## 1. Introduction

*Salmonella enterica* Enteritidis and Typhimurium, as well as *Campylobacter*, including *C. jejuni* and *C. coli*, are leading causes of bacterial-induced foodborne illness [1] with an economic burden in excess of 5.5 B USD per year in the United States (US) alone [2], and the number of culture-positive infections continues to increase each year [3]. In fact, the number of foodborne disease outbreaks associated with *Salmonella enterica* Enteritidis has maintained an increasing trend since 1965 [4] and continues to send people to the hospital [5]. Food production animals including cattle, swine, and chickens are known host reservoirs and transmission sources for both of these foodborne pathogens. According to the National Chicken Council, chicken is the number one protein consumed in the US and nearly 41,000 million pounds of ready-to-cook products were produced in 2017 [6]. Despite more stringent *Salmonella* and *Campylobacter* tolerance levels on processed carcasses [7], poultry products remain a source of foodborne illness. While the number of *Salmonella* outbreaks directly attributed to poultry has decreased [4], the same cannot be said for *Campylobacter* where estimates suggest poultry products still account for 50–70% of human campylobacteriosis cases [8]. A recent review eloquently covered the role of *Campylobacter* and *Salmonella* as causative agents of zoonotic foodborne disease, specifically addressing reservoirs for contamination, risk factors for contracting the disease, and the virulence mechanism(s) for each [9]. *Campylobacter* is “ubiquitous and adaptable” in broiler farms and once introduced can quickly colonize and spread through an entire flock [10]. As such, *Campylobacter* control will rely on a multifaceted approach ranging from enhanced biosecurity, feed/water treatments, litter amendments, and feed additives and yet may still only produce varied success. *Salmonella* control will rely on similar approaches but will also incorporate vaccination to control vertical transmission from the breeder flocks all the way to the broiler grown for meat consumption or in the hen laying table eggs [11]. Foodborne illness, in both instances, may result from the consumption of undercooked or from the improper handling of contaminated poultry products [12].

When discussing poultry and *Salmonella*, it is important to distinguish between the strains of *Salmonella* associated with foodborne disease outbreaks in humans and those responsible for the actual disease in the bird. As previously mentioned, *Salmonella enterica* Enteritidis and Typhimurium are causative agents associated with foodborne diseases. However, these two strains typically cause little to no disease in the bird. Nevertheless, two of the most important bacterial pathogens for the poultry industry are *Salmonella enterica* Pullorum and Gallinarum which are responsible for pullorum disease and fowl typhoid, respectively [13,14]. Even though these pathogens have been eradicated from flocks in the US, Canada, and most of Western Europe, they can be devastating to poultry industries in developing countries including China and may result in anorexia, diarrhea, dehydration, weakness, decreased egg production, poor fertility and hatchability, and mortality [15], which may be further complicated by the presence of antimicrobial resistant strains [16]. For the purpose of this review, discussions and intervention strategies will be directed towards controlling the *Salmonella* strains associated with foodborne illness which are not necessarily the appropriate interventions required to treat *Salmonella*-associated poultry diseases. 

The increased production and consumption of poultry has been largely led by the ability of the poultry industry to produce a quality protein in a relatively short timeframe while keeping the products affordable. In the past 50–60 years, there have been three key initiatives that primary poultry breeders have employed to meet the ever-changing demands of a global food market: (1) eradication of vertically transmitted diseases; (2) selection within and between genetic lines for increased livability and disease resistance; and (3) improved dissemination of husbandry practices and biosecurity [17]. Economic efficiency demanded by the poultry industry has directed the selection process towards fast-growing broiler chickens (*Gallus gallus*) with improved feed conversion ratios (FCR) and high yields. Such breeding has dramatically reduced the time required to get a bird to processing weight [17,18]. However, selection based heavily on growth characteristics and other phenotypic traits can adversely affect immune competence and leave birds more susceptible to diseases [19,20,21,22] and possibly leave them with a higher burden of key foodborne pathogens. 

In the US, traditional poultry management depends on the husbandry practices, biosecurity, vaccination, and when medically-necessary the application of broad-spectrum antibiotics. Historically, in addition to disease treatment, continual feeding of low-dose antibiotics as growth promoters (AGP) was standard practice [23]. However, much has changed in the past 10–15 years when the European Union (EU) withdrew approval for the use of growth-promoting, low-dose AGP in 2006 [24]. Studies suggest the misuse of antibiotics by all food animal industries is responsible, in part, for an increase in antibiotic-resistant bacteria. A recent review addressing antibiotic-resistant bacteria associated with the poultry industry, indicated that poultry may serve as source of resistant bacterial genes in zoonotic bacteria that present a real risk to human health [16]. This is supported by the findings of a recent study showing *Salmonella enterica* Enteritidis was the most commonly isolated serovar from US farms and that 7% of all isolates displayed a level of antibiotic resistance [25]. As a result of this and other factors, the general public is clamoring for removal of all drugs from poultry feed and has further pressured the poultry industry to find suitable alternative control and preventative measures [26]; therefore, research in this area has greatly increased. 

Simply defined, alternatives to antibiotics are “any substance that can be substituted for therapeutic drugs” either in their absence because of mandated withdrawal or reduced efficacy [27]. Identifying new approaches to manage flock health is becoming increasingly important as poultry breeding companies attempt to meet global consumption, fulfill consumer demands and comply with increased regulations all the while improving robustness, livability, production efficiency, and animal welfare. Therefore, the interaction(s) between the immune response and diet (immunomodulation) will be strategic areas of focus and research for controlling foodborne pathogens and improving animal health moving the poultry industry forward in the coming years. 

## 2. Immune Response

The host immune response to pathogens in the earliest stages of infection is a critical determinant of disease resistance and susceptibility over the life of a bird. There are two distinct divisions of an immune response referred to as innate and acquired immunity, and some of the key distinguishing characteristics associated with each are provided in Figure 1. Pattern recognition receptors (PRR) on host cells of the innate immune system recognize key pathogen or danger-associated molecular patterns (PAMPs or DAMPs) including polysaccharides, glycolipids, lipoproteins, nucleotides, and nucleic acids. This recognition by the innate immune system initiates the immediate defense against the threat and the interplay between the innate and adaptive immune responses is critical for coordinating the lasting response associated with adaptive immunity [28]. The coordinating intracellular signaling pathways can result in the activation of microbicidal killing mechanisms, the release of cytokines and chemokines, and the production of co-stimulatory molecules required for antigen presentation and activation of the acquired immune system [29,30]. It has been proposed that immune modulation targeting the innate system could prove more advantageous because induction is rapid, non-specific, yet offering therapeutic benefits including prophylaxis, adjuvant boosting effects, as well as local and systemic protection involving numerous cellular immune targets [27]. 

In chickens, some of the specific cells associated with an innate response are macrophages, heterophils, and B1-type lymphocytes that produce natural antibodies while B2 and T lymphocytes are more commonly associated with an acquired response [31]. Innate and acquired immunity work together to produce a response that is diverse and sensitive enough to recognize and eliminate a broad range of pathogens. The ultimate outcome of a successful immune response is for the bird to recognize the invading pathogen, prevent colonization, and eliminate the threat so as not to incur lasting negative affects to health and/or production. 

## 3. Nutrition

Nutrition encompasses more than merely feeding a bird a complete and balanced diet in today’s high-intensity production systems. Proper nutrition plays a vital role in enhancing productivity, helping to maintain a healthy gastrointestinal tract (GIT) and gut microflora, enabling the bird to reach its genetic potential, and improving animal health. With such a diverse impact on the bird, it is not surprising that nutrition can and does have the potential to either positively or negatively affect the host’s immune response. It is important to realize that these affects are largely due to the host–microbiota interaction(s) in the GIT where the microbiota process nutrients consumed by the host and then release metabolites from the intestinal lumen to the mucosal surface that directly impact host functions, including development, metabolism, and immunity [32,33]. Evaluation of specific feed ingredients and their impact on host immune function are outside the scope of this review since most interventions discussed herein are additive in nature and not specifically required for complete nutrition. However, this is an important aspect to animal health and some examples are provided elsewhere [31,34,35]. Additionally, there are outstanding reviews available that address the specific role of the GIT with regard to immunomodulation [27,36,37] and therefore, will not be covered herein. 

For the perspective of this review, the concept of modulation of the immune response will focus on the addition of feed additives to the dietary program to achieve a desired functional goal in the host (i.e., reduced colonization by foodborne pathogens and/or improved animal health). The literature is filled with published papers where investigators have evaluated an array of products as AGP alternatives including, but not limited to, food industry by-products, plant metabolites, non-digestible oligosaccharides, natural by-products, essential minerals, amino acids, medicinal herbs, organic acids, and essential oils. Another area of interest to reduce poultry pathogens as well as foodborne pathogens is centered around the use of pro and prebiotics, and because there are numerous studies in the literature addressing the role of these products in the poultry industry, we will only mention a few studies and recognize their importance. A recent review thoroughly addresses control strategies, including probiotics, in the control of *Campylobacter* in the poultry industry [38]. In other studies, feed supplementation with a prebiotic mixture altered the gut microflora and boosted the innate immune responsiveness in chickens challenged with *Salmonella enterica* Typhimurium [39] and using 55 probiotic bacterial strains showed some strains were able to persist in the chicken gut and reduce *Campylobacter jejuni* colonization and subsequent fecal shedding [40]. Clearly, modulation of the immune response is garnering much interest across all meat-producing sectors. 

## 4. Immunomodulation in Poultry

It has been said “Diets nourish immune cells, modulate them and facilitate establishment of commensal microflora, but diets shouldn’t normally stimulate the immune system. Let pathogens do that.” [31]. Clearly, there is a fine balance between when a diet simply nourishes the host and when stimulation occurs, and care must be taken to achieve the proper balance or bird health and/or performance could be compromised. However, continual feeding of an immunomodulatory product(s) throughout a broiler grow out does not necessarily mean the birds will be over-stimulated and experience poor performance. In fact, some findings report increased body weight (BW) and improved FCR with continual feeding. Broilers fed a phytogenic feed additive (PFA) report comparable BW gain and FCR to birds fed a diet containing AGP, suggesting such products are suitable alternatives to AGP in some instances [41]. In that study, the PFA-fed birds had a general increase in villus height through the small intestine and alterations in cecal microbiota composition that are typically associated with a healthy gut, suggesting the mechanism of action was through modulating the intestinal microbiota or by altering nutrient utilization in the GIT. These data demonstrate effective use of a PFA as a means to enhance animal health and welfare in the absence of AGP. The product administered in the above-mentioned study was proprietary, but some examples of PFAs that could be used for immunomodulatory purposes include essential oils, oleoresins, flavonoids, and bioactive molecules such as carvacrol, thymol, capsaicin, and cineole some of which are reviewed elsewhere [42,43]. To avoid the possible scenario of constant activation of the immune response and/or as a way to reduce expenses associated with the diet, the industry could consider implementing a strategy that specifically targets times when the birds are most susceptible to infections and/or other stressors. With respect to meat-type birds, times to consider immunomodulation could be in ovo, at hatch through the first week of life, at feed changes (starter to grower and grower to finisher), prior to transport to the processing facility, or during a disease outbreak. Additional evidence also indicates the diet fed to the broiler breeder can reduce chick mortality while increasing the immune responses indicating indirect modulation of the chick via the diet given to the broiler breeders [44]. 

During the first few days post-hatch, poultry are highly susceptible to pathogens many of which are commonly found in poultry facilities. In addition to the actual environmental sources, it is widely accepted that poultry feed can also serve as a direct source of contamination [45,46]. This susceptibility to infectious diseases can lead to significant economic losses and stems from an impairment of the innate and acquired host defense mechanisms. One of the most important features of this immune impairment is the functional inefficiency of the primary poultry polymorphonuclear cell (PMN), the heterophil. Functionally, heterophils are critical components of the innate immune response and exhibit an assortment of activities which include adhesion, chemotaxis, phagocytosis, production of cytokines/chemokines, and the microbicidal activities of degranulation and production of a respiratory burst [47]. Our laboratory has shown immunomodulation using prebiotics and antimicrobial peptides at the early post-hatch time was attainable and produced an immunologically superior bird compared to those given control, non-treated diets. Supplementation with a prebiotic such as beta-glucan [48], or antimicrobial peptides [49,50], resulted in enhanced heterophil function that translated to increased resistance against *Salmonella enterica* Enteritidis organ invasion and cecal colonization in young chicks. Additional studies on the impact of nutrition and feed additives on the innate immune response is reviewed elsewhere [51]. A number of the studies evaluating innate immunity lack performance data and additional experiments are needed to determine the true value of these approaches. However, the preliminary studies mentioned herein indicate select feed additives targeting the innate immune response may be used to successfully expedite and boost the development of early defense mechanisms to protect the young chicks. 

Foodborne and poultry pathogens are obvious concerns for the poultry industry and are becoming more difficult to manage between more stringent *Salmonella* and *Campylobacter* tolerance levels on processed carcasses [7]. It has also been reported that conventionally raised broilers have lower numbers of *Salmonella enterica* members compared to birds reared in antibiotic-free conditions [52]. Targeting and reducing the numbers of these two foodborne pathogens in the GIT of poultry will be key in reducing the incidence of foodborne illness acquired from handling or consuming contaminated or undercooked poultry. Successful delivery of a feed additive will depend, in part, on the stability of the compound being added and whether it can withstand the harsh environment of the GIT. One approach to circumvent this problem is to protect the compound by microencapsulating it with a lipid–embedding matrix [53], and once encapsulated, it is protected from the harsh environment, has improved stability, allows for a slow release, and thereby reduces the effective dose. By lowering the effective dose, producers can feed less compound, thus reducing cost. There are numerous studies in the literature clearly showing the benefit and effectiveness of such technology. In one such study, feeding a microencapsulated sorbic acid blend proved effective at reducing the load of *Salmonella enterica* Enteritidis by 2 log_10_ in market-age broilers following experimental challenge [54]. In a separate study, a different encapsulated product comprised of a blend of organic acids and botanicals similarly reduced the load of *Campylobacter jejuni* in slaughter-age broilers [55]. This finding is highly important as *Campylobacter jejuni* accounts for 80–85% of all infections associated with human gastroenteritis [12], further highlighting the significance of reducing *Campylobacter* burden in the live bird prior to processing. In addition to demonstrating the effectiveness of a microencapsulated product, these studies clearly show the potential for feed additives fed throughout or prior to transport to slaughter as a viable pre-harvest food safety intervention strategy to reduce the incidence of these highly important foodborne pathogens entering the food chain. Though important, laboratory challenges are not always comparable to field trials as shown in a study where a feed additive reduced *C. jejuni* colonization in broilers in laboratory trials but failed to reproduce those same beneficial effects under farm conditions [56]. This further supports what was stated earlier, that *Campylobacter* control will rely on a multifaceted approach ranging from enhanced biosecurity, feed/water treatment, litter amendments, and feed additives. 

## 5. Importance of Understanding the Mode-of-Action

If immunomodulation is to become a routine part of large-scale poultry production, it is imperative that research moving forward must begin to understand the mechanism(s) and mode-of-action behind the results. From an industry perspective, a product’s impact on performance and production traits will always be the driving factors as to whether a product is used or not. In most of the studies mentioned herein, the mechanism(s) resulting in the enhanced performance, reduced bacterial load, or increased resistance are not fully understood. 

There are several approaches to understanding the mode-of-action of any given feed additive. We have begun to dissect the mode-of-action of an encapsulated product containing citric acid, sorbic acid, thymol, and vanillin. The preliminary data showed feeding the encapsulated product the first four days post-hatch enhanced in vitro heterophil function including degranulation and oxidative burst (Swaggerty and Grilli, unpublished data), and earlier studies by our laboratory and others indicated increased heterophil function translated to increased resistance against *Salmonella enterica* Enteritidis [57,58,59] and *Campylobacter* [60]. Additional studies are required to fully understand the mode-of-action, but clearly, heterophil function should be considered as a contributing factor to the previously mentioned reduction in *Salmonella enterica* Enteritidis and Hadar [54]. *Salmonella enterica* Typhimurium uses inflammation products as a means to outgrow and outcompete other microbes in the intestine due to the reactive oxygen species that are generated during intestinal inflammation [61]. From a newer perspective, botanicals may act as crucial players in the chemosensory system by activating the transient receptor potential channels (TRP) [62]. TRP are expressed throughout the digestive tract and play several roles in taste, chemosensation, mechanosensation, pain, and control of motility by neurons [63]. They have been identified as mediators for the taste sensations of several spicy molecules [64] but more interestingly, recent advances in TRP research underlined their involvement in inflammation-mediated and immune-mediated diseases [65]. 

In order to clarify the mode-of-action of these additives, in addition to functional assays, newer technologies, such as a kinome array, are emerging as powerful research tools to dissect mechanisms. Kinome analysis using peptide arrays provide site-specific information, display similar biochemical properties to the full-length protein, and provide a means for defining phosphorylation-mediated events [66]. Phosphorylation is the predominant mechanism of post-translational modification regulating protein function, has a central role in virtually every cellular event, is essential for all cell signaling networks, and regulates fundamental biological processes [67,68]. Global analysis of the kinome provides information on the abundance, activity, substrate specificity, phosphorylation pattern, and mutational status of a given peptide [68], and chicken-specific kinome arrays are available [69,70]. Studies show the usefulness of kinome arrays to dissect key immuno-metabolic pathways associated with different feed additives. Therefore, making it possible to identify specific pathways and biomarkers associated with a given treatment compared to a control. One such study showed the addition of β-mannanase to broiler diets resulted in distinct changes in specific biological processes including regulation of immune response, innate immune response, MAPK signaling pathway, activation of immune response, and many others thereby providing mechanistic support for previously known outcomes associated with the feed additive [71]. Kinome studies can also provide insight while examining the complex interplay between the host and dietary supplementation under control and challenged conditions. Another study evaluated the effects of different preparations of yeast cell wall in non-challenged control birds compared to those experimentally infected with *C. perfringens* and found the level of purity of the feed additive impacted performance and could minimize the adverse impact of the challenge [72]. From a purely scientific standpoint, understanding the mechanism(s) is important, but even more so, this understanding will be vital for the poultry industry moving forward so they can make decisions based on sound science. 

## 6. Conclusions

The selected examples provided herein indicate immunomodulation can be achieved and is a practical means to enhance animal health and improve food safety without adversely impacting performance. To successfully raise commercial poultry in today’s setting, one must consider animal health and welfare and implement a multifaceted approach that keeps the bird free from harmful poultry pathogens all the while reducing the occurrence of foodborne pathogens. The role of immunomodulation is just one aspect of an integrated coordinated approach to produce a healthy bird that is also a safe and wholesome product for consumers.

## Figures and Tables

**Figure 1 microorganisms-07-00065-f001:**
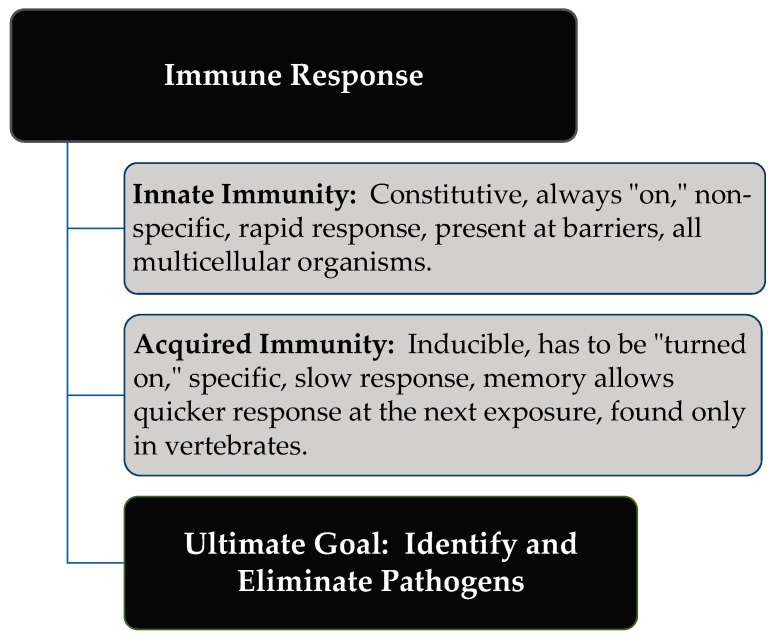
Two distinct divisions of an immune response: Innate and acquire immunity.

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
