# Peer review of "Modulation of the Immune Response to Improve Health and Reduce Foodborne Pathogens in Poultry"

_microorganisms, 2019, doi:10.3390/microorganisms7030065_

Round 1
Reviewer 1 Report
This is a very visionary review, which I enjoyed reading, sections 4 and 5 in particular. Its value goes far beyond poultry and I see perspectives linked to the present great interest of plant based food, in particular plant based fermented food, and human health. I recommend that the manuscript be accepted for publication after revision as indicated below.
The key poultry issue to be addressed in the review seems not to be clear. Is it “management of poultry flock health ” or “control of food borne human pathogens in poultry” ? This in particular applies for Salmonella serovars. Some Salmonella serovars are virulent to poultry but play no significant role as a food borne pathogens. For such serovars nutritional immune modulation may prevent disease in poultry but not necessarily control food borne human pathogens. This should be clarified. For Salmonella as well as Campylobacter genus names throughout the manuscript should be widely replaced by serovar or species names to indicate if poultry health or human food borne disease is addressed. The often pronounced differences in virulence and animal host reservoirs for Salmonella and Campylobacter in several cases also call for use of species rather than genus names.
The initiatives mentioned in Introduction to combat Salmonella and Campylobacter should be made specific and highlight the substantial differences applied in practice for the two genera of food borne pathogens. The importance of feed transmitted pathogens also deserves to be mentioned in the more general mentioning of combating pathogens.
It is seen that the review manuscript in particular addresses the US with statesments like traditional poultry management depends on application of broad-spectrum antibiotics. Readers from other parts of the world may prefer a different wording. It may also be fair to mention the problem of antibiotic resistance linked to use of antibiotics as feed additives to poultry.
I miss a more in depth analysis of the potential role of probiotic microorganisms in nutritional immune stimulation and as feed additive in poultry against Campylobacter jejuni. If possible not only in poultry but linked to fermented plant based feed. See also Santini et al. International Journal of Food Microbiology 141(2010) S98-S108.
For the role of nutrition and nutritional immuno-modulation (sections 3 and 4) several studies are mentioned, but a deeper analysis of the validity of the studies is not provided. E.g. , are the underlying intervention studies carried out as randomized, double–blinded cross-over trials? For Table 1 it is further suggested that references are provided specifically for the individual nutrients and their immune function. In other words are the effects mentioned science based? I did not locate a reference to Fig 1 in the main text, please check.
I just enjoyed reading sections 4 and 5.
Author Response
The key poultry issue to be addressed in the review seems not to be clear. Is it “management of poultry flock health” or “control of food borne human pathogens in poultry”? This in particular applies for Salmonella serovars. Some Salmonella serovars are virulent to poultry but play no significant role as a food borne pathogens. For such serovars nutritional immune modulation may prevent disease in poultry but not necessarily control food borne human pathogens. This should be clarified.
Yes, we apologize for not making this distinction clear for the reader as it is a critical point to make when discussing poultry and Salmonella. As such, the following has been added to the Introduction “When discussing poultry and Salmonella, it is important to distinguish between the strains of Salmonellaassociated with foodborne disease outbreaks in humans and those responsible for actual disease in the bird. As previously mentioned, Salmonellaenterica Enteritidis and Typhimurium are causative agents associated with foodborne disease; however, these two strains typically cause little to no disease in the bird. However, two of the most important bacterial pathogens for the poultry industry are Salmonella entericaPullorum and Gallinarum which are responsible for pullorum disease and fowl typhoid, respectively {Guo, 2018; Wigley, 2017}. Even though these pathogens have been eradicated from flocks in the US, Canada, and most of Western Europe, they can be devastating to poultry industries in developing countries including China and may result in anorexia, diarrhea, dehydration, weakness, decreased egg production, poor fertility and hatchability, and mortality {Barrow, 2011}, which may be further complicated by the presence of antimicrobial resistant strains {Nhung, 2017}. For the purpose of this review, discussions and intervention strategies will be directed towards controlling the Salmonellastrains associated with foodborne illness which are not necessarily the appropriate interventions required to treat Salmonella-associated poultry diseases.”
The authors also agree there is cross-over discussion between flock health and reducing the threat of foodborne pathogens, but we believe the approach of immunomodulation can and does have direct effects on both topics. To limit confusion and to demonstrate both areas are important, the title of the manuscript has been changed to “Modulation of the immune response to improve health and reduce foodborne pathogens in poultry.” Where appropriate terminology changes were incorporated into the manuscript.
For Salmonella as well as Campylobacter genus names throughout the manuscript should be widely replaced by serovar or species names to indicate if poultry health or human food borne disease is addressed. The often-pronounced differences in virulence and animal host reservoirs for Salmonella and Campylobacter in several cases also call for use of species rather than genus names.
Where appropriated the bacterial names were changed to include the serovar.
The initiatives mentioned in Introduction to combat Salmonella and Campylobacter should be made specific and highlight the substantial differences applied in practice for the two genera of food borne pathogens.
The following was added to the Introduction “A recent review eloquently covers the role of Campylobacterand Salmonella as causative agents of zoonotic foodborne disease specifically addressing reservoirs for contamination, risk factors for contracting the disease, and the virulence mechanism(s) for each {Chlebicz, 2018}. Campylobacteris “ubiquitous and adaptable” in broiler farms and once introduced can quickly colonize and spread through an entire flock {Wales, 2019}. As such, Campylobacter control will rely on a multifaceted approach ranging from enhanced biosecurity, feed/water treatment, litter amendments, and feed additives and yet may still only produce varied success. Salmonellacontrol will rely on similar approaches but will also incorporate vaccination to control vertical transmission from the breeder flocks all the way to the broiler grown for meat consumption or in the hen laying table eggs {Awad, 2014}.”
The importance of feed transmitted pathogens also deserves to be mentioned in the more general mentioning of combating pathogens.
The following statement was added “In addition to the actual environmental sources, it is widely accepted that poultry feed can also serve as a direct source of contamination {Jones, 2011; Magossi, 2018}.”
It is seen that the review manuscript in particular addresses the US with statements like traditional poultry management depends on application of broad-spectrum antibiotics. Readers from other parts of the world may prefer a different wording. It may also be fair to mention the problem of antibiotic resistance linked to use of antibiotics as feed additives to poultry.
For clarification of both comments, the paragraph was changed to “In the US, traditional poultry management depends on the husbandry practices, bio-security, vaccination, and when medically-necessary, application of broad-spectrum antibiotics. Historically, in addition to disease treatment, continual feeding of low dose antibiotics as growth promoters (AGP) was standard practice {Dibner, 2005}. However, much has changed in the past 10-15 years when the European Union (EU) withdrew approval for the use of AGP in 2006 {Castanon, 2007}. Studies suggest misuse of antibiotics by all food animal industries, is responsible, in part, for an increase in antibiotic-resistant bacteria. A recent review addressing antibiotic resistant bacteria associated with the poultry industry indicate poultry may serve as source of resistant bacterial genes in zoonotic bacteria that present a real risk to human health {Nhung, 2017}. This is supported by the findings of a recent study showing Salmonella entericaEnteritidis was the most commonly isolated serovar and that 7% of all isolates displayed a level of antibiotic resistance {Velasquez, 2018}. Further, the general public is clamoring for removal of all drugs from poultry feed and has further pressured the poultry industry to find suitable alternative control and preventative measures {McDougald, 1998}; therefore, research in this area has greatly increased.”
I miss a more in-depth analysis of the potential role of probiotic microorganisms in nutritional immune stimulation and as feed additive in poultry against Campylobacter jejuni. If possible not only in poultry but linked to fermented plant based feed. See also Santini et al. International Journal of Food Microbiology 141(2010) S98-S108.
The following was added to the manuscript “Another area of interest to reduce poultry pathogens as well as foodborne pathogens is centered around the use of pro-and prebiotics, and because there are numerous studies in the literature addressing the role of these products in the poultry industry, we will only mention a few studies and recognize their importance. A recent review thoroughly addresses control strategies, including probiotics, in the control of Campylobacterin the poultry industry {Meunier, 2016}. In other studies, feed supplementation with a prebiotic mixture altered the gut microflora and boosted the innate immune responsiveness in chickens challenged with Salmonella entericaTyphimurium {Faber, 2012} and using 55 probiotic bacterial strains showed some strains were able to persist in the chicken gut and reduce Campylobacter jejunicolonization and subsequent fecal shedding {Santini, 2010}.”
For the role of nutrition and nutritional immuno-modulation (sections 3 and 4) several studies are mentioned, but a deeper analysis of the validity of the studies is not provided. E.g. , are the underlying intervention studies carried out as randomized, double–blinded cross-over trials?
The studies that were referenced in these sections were all published in peer-reviewed journals and as such should have been properly vetted. However, the authors recognize that not all journals hold to rigorous standards, and for that reason, we are careful to include and reference studies that we believe show data generated under sound experimental design, that has been analyzed properly, and been reproduced. We don’t feel it necessary to include this comment in the text of the paper, but will certainly do so if you would like us to.
For Table 1 it is further suggested that references are provided specifically for the individual nutrients and their immune function. In other words are the effects mentioned science based? I did not locate a reference to Fig 1 in the main text, please check.
It was brought to our attention by another Reviewer that Table 1 was not specific nutrients and was quite diverse in the list provided. With that said, to limit any confusion Table 1 was removed from the paper and the following was added to the text “Evaluation of specific feed ingredients and their impact on host immune function are outside the scope of this review since most interventions discussed herein are additive in nature and not specifically required for complete nutrition; however this is an important aspect to animal health and some examples are provided elsewhere {Kidd, 2004;Klasing, 2007;Korver, 2012}.
Reviewer 2 Report
This paper presents a rather general but not very innovative literature review on the possible modulation of the immune response to foodborne pathogens in poultry induced by feed components. It suffers from a unbalanced view on some important aspects in the poultry industry.
In general, I wonder whether “nutritional (immuno)modulation” is well chosen here, because besides nutrients, also feed additives including fiber, bioactive molecules, essential oils, etc. are included which are not nutrients for the animal. With nutritional immunomodulation, I would understand an optimized or modified diet in terms of a changed dose or composition of the essential nutrients (protein, carbohydrates, fat, vitamins, minerals).
Another point is that the use of antibiotics in general, and of antimicrobial growth promotors (AGP’s) in specific, are linked in this paper with the control of foodborne pathogens such as Salmonella and Campylobacter, which is however not the case in practice. These pathogens are not treated with antibiotics as far as they cause no disease to the animals. The most important if not sole reason why the poultry industry is seeking for alternatives for antibiotics is the rise in antimicrobial resistance which is caused by the preventive/prophylactic or curative treatment of poultry against animal disease causing bacterial pathogens. One of these pathogens is Clostridium perfringens, which is described in the paper, but unfortunately not for its adequate importance in poultry. C. perfringens in poultry is causing necrotic enteritis in the animals, but this type of the pathogen is different from the one(s) causing foodborne disease in humans.
Antimicrobial growth promotors are or have not been used to manage flock health or treatment of disease, their intended purpose is to enhance growth performance by continuous dosing of antibiotics in the feed but at a much lower dose then for veterinary purposes. However, as a side-effect, they can have an effect on certain bacteria including pathogens as well; there are indications that the ban of AGP’s have led to an increase of C. perfringens infections in poultry.
Some other remarks are:
-L43: eggs are not associated with Campylobacter
-L62: there is no general ban of drugs in feed in the EU, as coccidiostats are still highly used in poultry feed.
-L123; table 1 does not present “specific nutrients” as it contains also very general nutrients such as protein, fiber,…Energy is not an nutrient
-L171-173: in line with the above major comments, there is no link between Salmonella and Campylobacter management and (reduced) availability of antibiotics.
-L182: how can a reduction of the effective dose be beneficial? Which are the numerous studies in literature?
-L184: not all foodborne pathogens are colonizing chickens early after hatch, this is certainly not the case for Campylobacter which usually appears only after 3-4 weeks when the maternal immunity in the chicks has disappeared.
-L187-195: only a few positive studies for Campylobacter reduction in poultry are given, while there are several studies showing no or only little effect especially in field studies.
-L227: what is this “abundant literature”? Where are at least some references or other reviews?
-L228-230: according to me, this sentence is contradictory formulated.
-L240-254: this paragraph would be more interesting if besides a general explanation of the kinome technology, it also gives more details what has already been obtained with it in the frame of mode of action of feed additives;
Author Response
In general, I wonder whether “nutritional (immuno)modulation” is well chosen here, because besides nutrients, also feed additives including fiber, bioactive molecules, essential oils, etc. are included which are not nutrients for the animal. With nutritional immunomodulation, I would understand an optimized or modified diet in terms of a changed dose or composition of the essential nutrients (protein, carbohydrates, fat, vitamins, minerals).
The authors agree and have changed the title of the manuscript to “Modulation of the immune response to improve health and reduce foodborne pathogens in poultry.” Where appropriate changes were incorporated into the manuscript.
Another point is that the use of antibiotics in general, and of antimicrobial growth promotors (AGP’s) in specific, are linked in this paper with the control of foodborne pathogens such as Salmonella and Campylobacter, which is however not the case in practice. These pathogens are not treated with antibiotics as far as they cause no disease to the animals.
When discussing poultry and Salmonella, it is important to distinguish between the strains of Salmonellaassociated with foodborne disease outbreaks in humans and those responsible for actual disease in the bird. As previously mentioned, Salmonellaenterica Enteritidis and Typhimurium are causative agents associated with foodborne disease; however, these two strains typically cause little to no disease in the bird. However, two of the most important bacterial pathogens for the poultry industry are Salmonella entericaPullorum and Gallinarum which are responsible for pullorum disease and fowl typhoid, respectively {Guo, 2018; Wigley, 2017}. Even though these pathogens have been eradicated from flocks in the US, Canada, and most of Western Europe, they can be devastating to poultry industries in developing countries including China and may result in anorexia, diarrhea, dehydration, weakness, decreased egg production, poor fertility and hatchability, and mortality {Barrow, 2011}, which may be further complicated by the presence of antimicrobial resistant strains {Nhung, 2017}. For the purpose of this review, discussions and intervention strategies will be directed towards controlling the Salmonellastrains associated with foodborne illness which are not necessarily the appropriate interventions required to treat Salmonella-associated poultry diseases.
The most important if not sole reason why the poultry industry is seeking for alternatives for antibiotics is the rise in antimicrobial resistance which is caused by the preventive/prophylactic or curative treatment of poultry against animal disease causing bacterial pathogens.
To address your comments regarding AGP and resistance the following was added to the text “In the US, traditional poultry management depends on the husbandry practices, bio-security, vaccination, and when medically-necessary, application of broad-spectrum antibiotics. Historically, in addition to disease treatment, continual feeding of low-dose antibiotics as growth promoters (AGP) was standard practice {Dibner, 2005}. However, much has changed in the past 10-15 years when the European Union (EU) withdrew approval for the use of growth-promoting low-dose AGP in 2006 {Castanon, 2007}. Studies suggest misuse of antibiotics by all food animal industries, is responsible, in part, for an increase in antibiotic-resistant bacteria. A recent review addressing antibiotic resistant bacteria associated with the poultry industry indicate poultry may serve as source of resistant bacterial genes in zoonotic bacteria that present a real risk to human health {Nhung, 2017}. This is supported by the findings of a recent study showing Salmonella entericaEnteritidis was the most commonly isolated serovar from US farms and that 7% of all isolates displayed a level of antibiotic resistance {Velasquez, 2018}. Because of this and other factors, the general public is clamoring for removal of all drugs from poultry feed and has further pressured the poultry industry to find suitable alternative control and preventative measures {McDougald, 1998}; therefore, research in this area has greatly increased.”
One of these pathogens is Clostridium perfringens, which is described in the paper, but unfortunately not for its adequate importance in poultry. C. perfringens in poultry is causing necrotic enteritis in the animals, but this type of the pathogen is different from the one(s) causing foodborne disease in humans.
With respect to your Clostridium comment, the followingwas added to the text for clarification “These data suggest feeding bacteriocins during an outbreak with NE could potentially reduce the losses associated with this very costly and important poultry pathogen. Further, gut damage can leave birds more susceptible to human pathogens including Salmonella{Volkova, 2011}. So, if NE-associated gut damage is reduced, there would be less opportunity for Salmonella,and possibly other foodborne pathogens, to colonize the chicken GIT therefore reducing the chance that these pathogens could enter the food chain.”
Antimicrobial growth promotors are or have not been used to manage flock health or treatment of disease, their intended purpose is to enhance growth performance by continuous dosing of antibiotics in the feed but at a much lower dose then for veterinary purposes. However, as a side-effect, they can have an effect on certain bacteria including pathogens as well; there are indications that the ban of AGP’s have led to an increase of C. perfringens infections in poultry.
The authors agree with these comments. For clarification “continual feeding of low dose antibiotics” was included Section 1, paragraph 3. In fact, we mentioned in the original text (L201-203) that “Removal of AGP from poultry diets has resulted in an increased incidence of coccidiosis and Clostridium-induced NE and gangrenous dermatitis in the broiler industry {Abbas, 2011;Gaucher, 2015}.”
L43: eggs are not associated with Campylobacter
The sentence was changed to “Foodborne illness, in both instances, may result from consumption of undercooked or improper handling of contaminated poultry products.”
L62: there is no general ban of drugs in feed in the EU, as coccidiostats are still highly used in poultry feed.
For clarification, the sentence was changed to “However, much has changed in the past 10-15 years when the European Union (EU) withdrew approval for the use of growth-promoting, low-dose AGP in 2006 {Castanon, 2007}.”
L123; table 1 does not present “specific nutrients” as it contains also very general nutrients such as protein, fiber,…Energy is not an nutrient
The authors agree and to limit any confusion Table 1 was removed from the paper and the following was added to the text “Evaluation of specific feed ingredients and their impact on host immune function are outside the scope of this review since most interventions discussed herein are additive in nature and not specifically required for complete nutrition; however this is an important aspect to animal health and some examples are provided elsewhere {Kidd, 2004;Klasing, 2007;Korver, 2012}.
L171-173: in line with the above major comments, there is no link between Salmonella and Campylobacter management and (reduced) availability of antibiotics.
The authors agree and the sentence was changed to “Foodborne and poultry pathogens are obvious concerns for the poultry industry and are becoming more difficult to manage between more stringent Salmonellaand Campylobactertolerance levels on processed carcasses.”
L182: how can a reduction of the effective dose be beneficial? Which are the numerous studies in literature?
For clarification, the following was added to the text “By lowering the effective dose, producers can feed less compound, thus reducing cost.” The statement “There are numerous studies in the literature clearly showing the benefit and effectiveness of such technology.” was removed from the text. However, the studies the authors were eluding to are described in more detail in the following two paragraphs and include references 42, 43, and 48 in the original submission.
L184: not all foodborne pathogens are colonizing chickens early after hatch, this is certainly not the case for Campylobacter which usually appears only after 3-4 weeks when the maternal immunity in the chicks has disappeared.
The sentence was changed to “Chickens are typically colonized by foodborne pathogens within the first 3 wk post-hatch and most remain carriers for life, and it is this carriage state that increases the likelihood of carcass contamination during slaughter and processing.”
L187-195: only a few positive studies for Campylobacter reduction in poultry are given, while there are several studies showing no or only little effect especially in field studies.
We will gladly address your comment/concern, but do not understand what you are asking in this comment.
L227: what is this “abundant literature”? Where are at least some references or other reviews?
The following references were added and the statement now reads “With respect to thymol, and more in general monoterpenes and botanicals, there are numerous studies describing their antioxidant activity to explain the anti-inflammatory mode of action {Marsik, 2005; Landa, 2009; Meeran, 2017}.”
L228-230: according to me, this sentence is contradictory formulated.
The sentence was deleted.
L240-254: this paragraph would be more interesting if besides a general explanation of the kinome technology, it also gives more details what has already been obtained with it in the frame of mode of action of feed additives.
The following was added to the paragraph “Studies show the usefulness of kinome arrays to dissect key immuno-metabolic pathways associated with different feed additives; therefore, making it possible to identify specific pathways and biomarkers associated with a given treatment compared to a control. One such study shows the addition of b-mannanase to broiler diets results in distinct changes in specific biological processes including regulation of immune response, innate immune response, MAPK signaling pathway, activation of immune response, and many others thereby providing mechanistic support for previously known outcomes associated with the feed additive {Arsenault, 2017}. Kinome studies can also provide insight while examining the complex interplay between the host and dietary supplementation under control and challenged conditions. Hashim and colleagues evaluated the effects of different preparations of yeast cell wall in non-challenged control birds compared to those experimentally-infected with C. perfringensand found the level of purity of the feed additive impacts performance and may minimize the adverse impact of the challenge {Hashim, 2018}. From a purely scientific standpoint, understanding mechanism(s) is important, but even more so, this understanding will be vital for the poultry industry moving forward so they can make decisions based on sound science.”
Reviewer 3 Report
This is a review article, so many of the questions above do not apply (research design, methodology, etc). It is very well written and sourced, and I believe will be of great interest to researchers in food safety-related fields.
Author Response
This is a review article, so many of the questions above do not apply (research design, methodology, etc). It is very well written and sourced, and I believe will be of great interest to researchers in food safety-related fields.
The authors thank the Reviewer for their time and effort in reviewing our paper and appreciated the kind comments.
Round 2
Reviewer 2 Report
The paper has improved to some extent by focusing on foodborne pathogens and Salmonella and Campylobacter in particular in poultry.
But some major remarks remain.
The major comment about the terminology “nutritional immunomodulation” remains to be handled consistently in the manuscript. Now it seems only to be changed in the title. The main problem is that within such a concept bioactive components like pre- and probiotics and botanicals are being considered as nutrients while they are additives to a normal diet. The manuscript is very blurry in this aspect. A better concept then “nutritional immunomodulation”, and of “nutritional” in specific, would be very helpful to clarify things.
The paragraph on C. perfringens, is still about a poultry pathogen, not about a foodborne pathogen. According to me, this is out of the scope of this review. The statement that gut damage would leave the birds more susceptible to human pathogens is based on a reference on protozoal and viral infections, and so the link with NE-associated gut damage, is speculative and not based on primary references.
Some other remarks:
-L238: the typical colonization of chickens with foodborne pathogens during the first 3 weeks, is not valid for Campylobacter as already mentioned in my comment before. As Campylobacter is the major foodborne pathogen from chickens, the authors should state this more correctly by looking for the appropriate literature references.
-L244-246: as also commented before, a lot of in vivo challenge studies have shown promising results for reducing Campylobacter excretion in chickens, but the main issue which remains to be answered in this review, is whether these promising studies like the one given (ref. 56), have been validated in field conditions using naturally contaminated animals. As far as I know, there is still not a valid product on the market or from research to tackle this pathogen. I belief it is important to give the reader an appropriate evaluation.
Author Response
We would like to thank the reviewer for your detailed reading and evaluation of this manuscript. The constructive criticism provided has strengthened the manuscript and enhanced the science. We have tried to make every change, and clarify what we could not change to meet the reviewers outstanding suggestions.
The major comment about the terminology “nutritional immunomodulation” remains to be handled consistently in the manuscript. Now it seems only to be changed in the title. The main problem is that within such a concept bioactive components like pre- and probiotics and botanicals are being considered as nutrients while they are additives to a normal diet. The manuscript is very blurry in this aspect. A better concept then “nutritional immunomodulation”, and of “nutritional” in specific, would be very helpful to clarify things.
The authors have carefully worked through the manuscript and where appropriate have removed the terms nutrition, nutritional, or similar words, to minimize the confusion. We hope the Reviewer agrees and can see that we are speaking of modulation of the immune response without focusing on specific nutrients and dietary make-up, etc. as a means to do so. We believe Section 3 on Nutrition is useful as written and clarifies for the reader what we are addressing in the review “For the perspective of this short review, the concept of modulation of the immune response will focus on the addition of feed additives to the dietary program to achieve a desired functional goal in the host (i.e. reduced colonization by foodborne pathogens and/or improved animal health).”
The paragraph on C. perfringens, is still about a poultry pathogen, not about a foodborne pathogen. According to me, this is out of the scope of this review. The statement that gut damage would leave the birds more susceptible to human pathogens is based on a reference on protozoal and viral infections, and so the link with NE-associated gut damage, is speculative and not based on primary references.
The authors agree that the Clostridium paragraph was outside the scope of the review and removed it.
L238: the typical colonization of chickens with foodborne pathogens during the first 3 weeks, is not valid for Campylobacter as already mentioned in my comment before. As Campylobacter is the major foodborne pathogen from chickens, the authors should state this more correctly by looking for the appropriate literature references.
The sentence in question was removed (Chickens are typically colonized by foodborne pathogens within the first 3 wk post-hatch and most remain carriers for life, and it is this carriage state that increases the likelihood of carcass contamination during slaughter and processing [54].) and the paragraph was com bined with the one prior to it. The sentence that was removed was just meant as a transition, so removing it was the appropriate action.
L244-246: as also commented before, a lot of in vivo challenge studies have shown promising results for reducing Campylobacter excretion in chickens, but the main issue which remains to be answered in this review, is whether these promising studies like the one given (ref. 56), have been validated in field conditions using naturally contaminated animals. As far as I know, there is still not a valid product on the market or from research to tackle this pathogen. I belief it is important to give the reader an appropriate evaluation.
For clarification, the following was added “Though important, laboratory challenges are not always comparable to field trials as shown in a study where a feed additive reduced C. jejunicolonization in broilers in laboratory trials but failed to reproduce those same beneficial effects under farm conditions {Huneau-Salaun, 2018}. This further supports what was stated earlier, that Campylobactercontrol will rely on a multifaceted approach ranging from enhanced biosecurity, feed/water treatment, litter amendments, and feed additives.”